# Paths-over-Graph: Knowledge Graph Enpowered Large Language Model Reasoning

## ABSTRACT

Large Language Models (LLMs) have achieved impressive results in various tasks but struggle with hallucination problems and lack of relevant knowledge, especially in deep complex reasoning and knowledge-intensive tasks. Knowledge Graphs (KGs), which capture vast amounts of facts in a structured format, offer a reliable source of knowledge for reasoning. However, existing KG-based LLM reasoning methods face challenges like handling multi-hop reasoning, multi-entity questions, and effectively utilizing graph structures. To address these issues, we propose Paths-over-Graph (PoG), a novel method that enhances LLM reasoning by integrating knowledge reasoning paths from KGs, improving the interpretability and faithfulness of LLM outputs. PoG tackles multi-hop and multi-entity questions through a three-phase dynamic multi-hop path exploration, which combines the inherent knowledge of LLMs with factual knowledge from KGs. In order to improve the efficiency, PoG prunes irrelevant information from the graph exploration first and introduces efficient three-step pruning techniques that incorporate graph structures, LLM prompting, and a pre-trained language model (e.g., SBERT) to effectively narrow down the explored candidate paths. This ensures all reasoning paths contain highly relevant information captured from KGs, making the reasoning faithful and interpretable in problem-solving. PoG innovatively utilizes graph structure to prune the irrelevant noise and represents the first method to implement multi-entity deep path detection on KGs for LLM reasoning tasks. Comprehensive experiments on five benchmark KGQA datasets demonstrate PoG outperforms the state-of-the-art method ToG across GPT-3.5-Turbo and GPT-4, achieving an average accuracy improvement of 18.9%. Notably, PoG with GPT-3.5-Turbo surpasses ToG with GPT-4 by up to 23.9%.

**ACM Reference Format:**
Anonymous Author(s). 2018. Paths-over-Graph: Knowledge Graph Enpowered Large Language Model Reasoning. In *Proceedings of Make sure to enter the correct conference title from your rights confirmation emai (Conference acronym 'XX).* ACM, New York, NY, USA, 17 pages. https://doi.org/XXXXXXX.XXXXXXX

## 1 INTRODUCTION

Large Language Models (LLMs) have demonstrated remarkable performance in various tasks [4, 6, 8, 37]. These models leverage pre-training techniques by scaling to billions of parameters and training

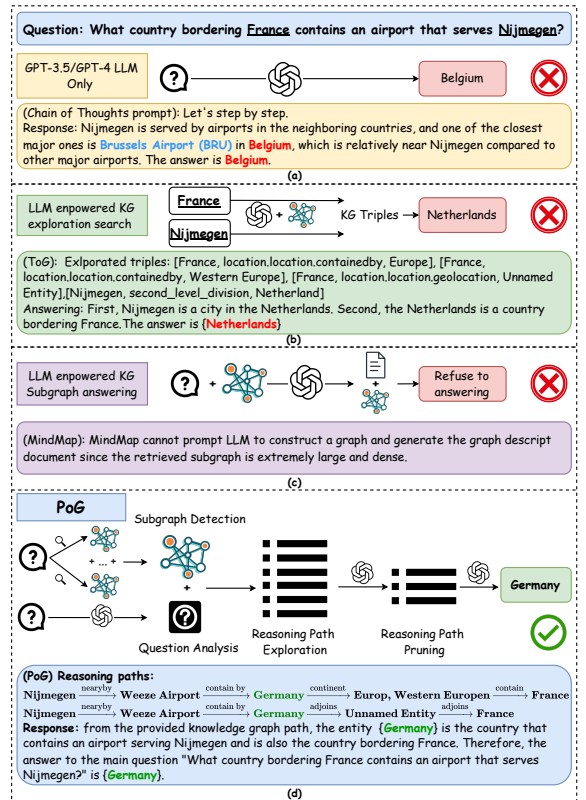

**Figure 1: Representative workflow of four LLM reasoning paradigms**

on extensive, diverse, and unlabeled data [31, 37]. Despite these impressive capabilities, LLMs face two well-known challenges. First, they struggle with deep and responsible reasoning when tackling complex tasks [19, 30, 36]. Second, the substantial cost of training makes it difficult to keep models updated with the latest knowledge [34, 41], leading to errors when answering questions that require specialized information not included in their training data. For example, in Figure 1(a), though models like GPT can generate reasonable answers for knowledge-specific questions, these answers may be incorrect due to outdated information or hallucination of reasoning on LLMs inherent Knowledge Base (KB).

To deal with the problems of error reasoning and knowledge gaps, the `plan-retrieval-answering` method has been proposed [23, 25, 48]. In this approach, LLMs are prompted to decompose complex reasoning tasks into a series of sub-tasks, forming a `plan`. Simultaneously, external KBs are retrieved to answer each step of the plan. However, this method still has the issue of heavily relying on the reasoning abilities of LLMs rather than the faithfulness of the retrieved knowledge. The generated reasoning steps guide

information selection, but answers are chosen based on the LLM's interpretation of the retrieved knowledge rather than on whether the selection leads to a correct and faithful answer.

To address these challenges, incorporating external knowledge sources like Knowledge Graphs (KGs) is a promising solution to enhance LLM reasoning [24, 25, 27, 34]. KGs offer abundant factual knowledge in a structured format, serving as a reliable source to improve LLM capabilities. Knowledge Graph Question Answering (KGQA) serves as an approach for evaluating the integration of KGs with LLMs, which requires machines to answer natural language questions by retrieving relevant facts from KGs. These approaches typically involve: (1) identifying the initial entities from the question, and (2) iteratively retrieving and refining inference paths until sufficient evidence has been obtained. Despite their success, they still face challenges such as handling multi-hop reasoning problems, addressing questions with multiple topic entities, and effectively utilizing the structural information of graphs.

Challenge 1: Multi-hop reasoning problem. Current methods [16, 26, 34, 44], such as the ToG model presented in Figure 1(b), begin by exploring from each topic entity, with LLMs selecting connected knowledge triples like (France, contained_by, Europe). This process relies on the LLM's inherent understanding of these triples. However, focusing on one-hop neighbors can result in plausible but incorrect answers and prematurely exclude correct ones, especially when multi-hop reasoning is required. Additionally, multi-hop reasoning introduces significant computational overhead, making efficient pruning essential, especially in dense and large KGs.

Challenge 2: Multi-entity question. As shown in Figure 1(b), existing work [16, 26, 34, 44] typically explores KG for each topic entity independently. When a question involves multiple entities, these entities are examined in separate steps without considering their interconnections. This approach can result in a large amount of irrelevant information in the candidate set that does not connect to the other entities in the question, leading to suboptimal results.

Challenge 3: Utilizing graph structure. Existing methods [7, 14, 41] often overlook the inherent graph structures when processing retrieved subgraphs. For example, the MindMap model in Figure 1(c) utilizes LLMs to generate text-formatted subgraphs from KG triples, converting them into graph descriptions that are fed back into the LLM to produce answers. This textual approach overlooks the inherent structural information of graphs and can overwhelm the LLM when dealing with large graphs. Additionally, during KG information selection, most methods use in-context learning by feeding triples into the LLM, ignoring the overall graph structure.

**Contributions**. In this paper, we introduce a novel method, **P**aths-**o**ver-**G**raph (**PoG**). Unlike previous studies that utilize knowledge triples for retrieval [26, 34], PoG employs knowledge reasoning paths, that contained all the topic entities in a long reasoning length, as a retrieval-augmented input for LLMs. The paths in KGs serve as logical reasoning chains, providing KG-supported, interpretable reasoning logic that addresses issues related to the lack of specific knowledge background and unfaithful reasoning paths.

To address multi-hop reasoning problem, as shown in Figure 1(d), PoG first performs question analysis, to extract topic entities from questions. Utilizing these topic entities, it decomposes the complex question into sub-questions and generates an LLM thinking indicator termed "Planning". This planning not only serves as an answering strategy but also predicts the implied relationship depths between the answer and each topic entity. The multi-hop paths are then explored starting from a predicted depth, enabling a dynamic search process. Previous approaches using planing usually retrieve information from scratch, which often confuse LLMs with source neighborhood-based semantic information. In contrast, our method ensures that LLMs follow accurate reasoning paths that directly lead to the answer.

To address multi-entity questions, PoG employs a three-phase exploration process to traverse reasoning paths from the retrieved question subgraph. All paths must contain all topic entities in the same order as they occur in the LLM thinking indicator. In terms of reasoning paths in KGs, all paths are inherently logical and faithful. Each path potentially contains one possible answer and serves as the interpretable reasoning logic. The exploration leverages the inherent knowledge of both LLM and KG.

To effectively utilize graph structure, PoG captures the question subgraph by expanding topic entities to their maximal depth neighbors, applying graph clustering and reduction to reduce graph search cost. In the path pruning phase, we select possible correct answers from numerous candidates. All explored paths undergo a three-step beam search pruning, integrating graph structures, LLM prompting, and a pre-trained language understanding model (e.g., BERT) to ensure effectiveness and efficiency. Additionally, inspired by the Graph of Thought (GoT) [4], to reduce LLM hallucination, PoG prompts LLMs to summarize the obtained Top-$W_{max}$ paths before evaluating the answer, where $W_{max}$ is a user-defined maximum width in the path pruning phase. In summary, the advantage of PoG can be abbreviated as:

- **Dynamic deep search**: Guided by LLMs, PoG dynamically extracts multi-hop reasoning paths from KGs, enhancing LLM capabilities in complex knowledge-intensive tasks.
- **Interpretable and faithful reasoning**: By utilizing highly question-relevant knowledge paths, PoG improves the interpretability of LLM reasoning, enhancing the faithfulness and question-relatedness of LLMs-generated content.
- **Efficient pruning with graph structure integration**: PoG incorporates efficient pruning techniques in both the KG and reasoning paths to reduce computational costs, mitigate LLM hallucinations caused by irrelevant noise, and effectively narrow down candidate answers.
- **Flexibility and effectiveness**: a) PoG is a plug-and-play framework that can be seamlessly applied to various LLMs and KGs. b) PoG allows frequent knowledge updates via the KG, avoiding the expensive and slow updates required for LLMs. c) PoG reduces the LLMs token usage by over 50% with only a ±2% difference in accuracy compared to the best-performing strategy. d) PoG achieves state-of-the-art results on all the tested KGQA datasets, outperforming the strong baseline ToG by an average of 18.9% accuracy using both GPT-3.5 and GPT-4. Notably, PoG with GPT-3.5 can outperform ToG with GPT-4 by up to 23.9%.

## 2 RELATED WORK

**KG-based LLM reasoning**. KGs provide structured knowledge valuable for integration with LLMs [27]. Early studies [23, 25, 28, 46]

embed KG knowledge into neural networks during pre-training or fine-tuning, but this can reduce explainability and hinder efficient knowledge updating [27]. Recent methods combine KGs with LLMs by converting relevant knowledge into textual prompts, often ignoring structural information [27, 41]. Advanced works [17, 26, 34] involve LLMs directly exploring KGs, starting from an initial entity and iteratively retrieving and refining reasoning paths until the LLM decides the augmented knowledge is sufficient. However, by starting from a single vertex and ignoring the question's position within the KG's structure, these methods overlook multiple topic entities and the explainability provided by multi-entity paths.

**Reasoning with LLM prompting**. LLMs have shown significant potential in solving complex tasks through effective prompting strategies. Chain of Thought (CoT) prompting [40] enhances reasoning by following logical steps in few-shot learning. Extensions like Auto-CoT [47], Complex-CoT [10], CoT-SC [39], Zero-Shot CoT [21], ToT [42], and GoT [4] build upon this approach. However, these methods often rely solely on knowledge present in training data, limiting their ability to handle knowledge-intensive or deep reasoning tasks. To solve this, some studies integrate external KBs using plan-and-retrieval methods such as CoK [23], RoG [25], and ReAct [43], decomposing complex questions into subtasks to reduce hallucinations. However, they may focus on the initial steps of sub-problems and overlook further steps of final answers, leading to locally optimal solutions instead of globally optimal ones. To address these deep reasoning challenges, we introduce dynamic multi-hop question reasoning. By adaptively determining reasoning depths for different questions, we enable the model to handle varying complexities effectively.

**KG information pruning**. KGs contain vast amounts of facts [15], making it impractical to involve all relevant triples in the context of the LLM due to high costs and potential noise [38]. Existing methods [17, 26, 34] typically identify initial entities and iteratively retrieve reasoning paths until an answer is reached, often treating the LLM as a function executor and relying on in-context learning or fine-tuning, which is expensive. Some works attempt to reduce pruning costs. KAPING [2] projects questions and triples into the same semantic space to retrieve relevant knowledge via similarity measures. KG-GPT [20] decomposes multi-hop questions into sub-questions, matches entity relations, and selects top-$k$ relevant relations to form evidence triples. Similarly, KGR [13] splits retrieved triples into chunks and uses LLMs to identify critical ones. However, these methods often overlook the overall graph structure and the interrelations among multiple topic entities, leading to suboptimal pruning and reasoning performance.

## 3 PRELIMINARY

Consider a Knowlegde Graph (KG) $\mathcal{G}(\mathcal{E}, \mathcal{R}, \mathcal{T})$, where $\mathcal{E}$, $\mathcal{R}$ and $\mathcal{T}$ represent the set of entities, relations, and knowledge triples, respectively. Each knowledge triple $T \in \mathcal{T}$ encapsulates the factual knowledge in $\mathcal{G}$, and is represented as $T = (e_h, r, e_t)$, where $e_h, e_t \in \mathcal{E}$ and $r \in \mathcal{R}$. Given an entity set $\mathcal{E}_S \subseteq \mathcal{E}$, the induced subgraph of $\mathcal{E}_S$ is denoted as $\mathcal{S} = (\mathcal{E}_S, \mathcal{R}_S, \mathcal{T}_S)$, where $\mathcal{T}_S = \{(e, r, e') \in \mathcal{T} \mid e, e' \in \mathcal{E}_S\}$, and $\mathcal{R}_S = \{r \in \mathcal{R} \mid (e, r, e') \in \mathcal{T}_S\}$. Furthermore, we denote $\mathcal{D}(e)$ and $\mathcal{D}(r)$ as the sets of short textual descriptions for each entity $e \in \mathcal{E}$ and each relation $r \in \mathcal{R}$, respectively. For example, the text description of the entity "m.0f8l9c" is $\mathcal{D}(\text{"m.0f8l9c"}) =$ "France". For simplicity, in this paper, all entities and relations are referenced through their $\mathcal{D}$ representations and transformed into natural language.

**DEFINITION 1 (REASONING PATH).** *Given a KG $\mathcal{G}$, a reasoning path within $\mathcal{G}$ is defined as a connected sequence of knowledge triples, represented as: $path_{\mathcal{G}}(e_1, e_{l+1}) = \{T_1, T_2, ..., T_l\} = \{(e_1, r_1, e_2), (e_2, r_2, e_3), ..., (e_l, r_l, e_{l+1})\}$, where $T_i \in \mathcal{T}$ denotes the i-th triple in the path and l denotes the length of the path, i.e., $length(path_{\mathcal{G}}(e_1, e_{l+1})) = l$.*

**EXAMPLE 1.** *Consider a reasoning path between the entity "University" and the entity "Student" in a KG. The reasoning path is given by: $path_{\mathcal{G}}(University, Student) = \{(University, employs, Professor), (Professor, teaches, Course), (Course, enrolled\_in, Student)\}$, and can be visualized as:*

$$University \xrightarrow{employs} Professor \xrightarrow{teaches} Course \xrightarrow{enrolled\_in} Student.$$

*This path indicates that a "University" employs a "Professor," who teaches a "Course," in which a "Student" is enrolled. The length of the path is 3.*

For any entity $s$ and $t$ in $\mathcal{G}$, if there exists a reasoning path between $s$ and $t$, we say $s$ and $t$ can reach each other, denoted as $s \leftrightarrow t$. The distance between $s$ and $t$ in $\mathcal{G}$, denoted as $dist_{\mathcal{G}}(s, t)$, is the shortest reasoning path distance between $s$ and $t$. For the non-reachable vertices, their distance is infinite. Given a positive integer $h$, the $h$-hop neighbors of an entity $s$ in $\mathcal{G}$ is defined as $N_{\mathcal{G}}(s, h) = \{t \in \mathcal{E} | dist_{\mathcal{G}}(s, t) \leq h\}$.

**DEFINITION 2 (ENTITY PATH).** *Given a KG $\mathcal{G}$ and a list of entities $list_e = [e_1, e_2, e_3, \ldots, e_l]$, the entity path of $list_e$ is defined as a connected sequence of reasoning paths, which is denoted as $path_{\mathcal{G}}(list_e) = \{path_{\mathcal{G}}(e_1, e_2), path_{\mathcal{G}}(e_2, e_3), \ldots, path_{\mathcal{G}}(e_{l-1}, e_l)\} = \{(e_s, r, e_t) | (e_s, r, e_t) \in path_{\mathcal{G}}(e_i, e_{i+1}) \wedge 1 \leq i < l\}.$*

Knowledge Graph Question Answering (KGQA) is a fundamental reasoning task based on KGs. Given a natural language question $q$ and a KG $\mathcal{G}$, the objective is to devise a function $f$ that predicts answers $a \in Answer(q)$ utilizing knowledge encapsulated in $\mathcal{G}$, i.e., $a = f(q, \mathcal{G})$. Consistent with previous research [25, 26, 33, 34], we assume the topic entities $Topic(q)$ mentioned in $q$ and answer entities $Answer(q)$ in ground truth are linked to the corresponding entities in $\mathcal{G}$, i.e., $Topic(q) \subseteq \mathcal{E}$ and $Answer(q) \subseteq \mathcal{E}$.

## 4 METHOD

PoG implements the "KG-based LLM Reasoning" by first exploring all possible faithful reasoning paths and then collaborating with LLM to perform a 3-step beam search selection on the retrieved paths. Compared to previous approaches [26, 34], our model focuses on providing more accurate and question-relevant retrieval-argument graph information. The framework of PoG is outlined in Figure 2, comprising four main components.

- **Initialization.** The process begins by identifying the set of topic entities from the question input, and then queries the source KG $\mathcal{G}$ by exploring up to $D_{\max}$-hop from each topic entity to construct the evidence sub-graph $\mathcal{G}_q$, where $D_{\max}$ is the user-defined maximum exploration depth. Subsequently, we prompt

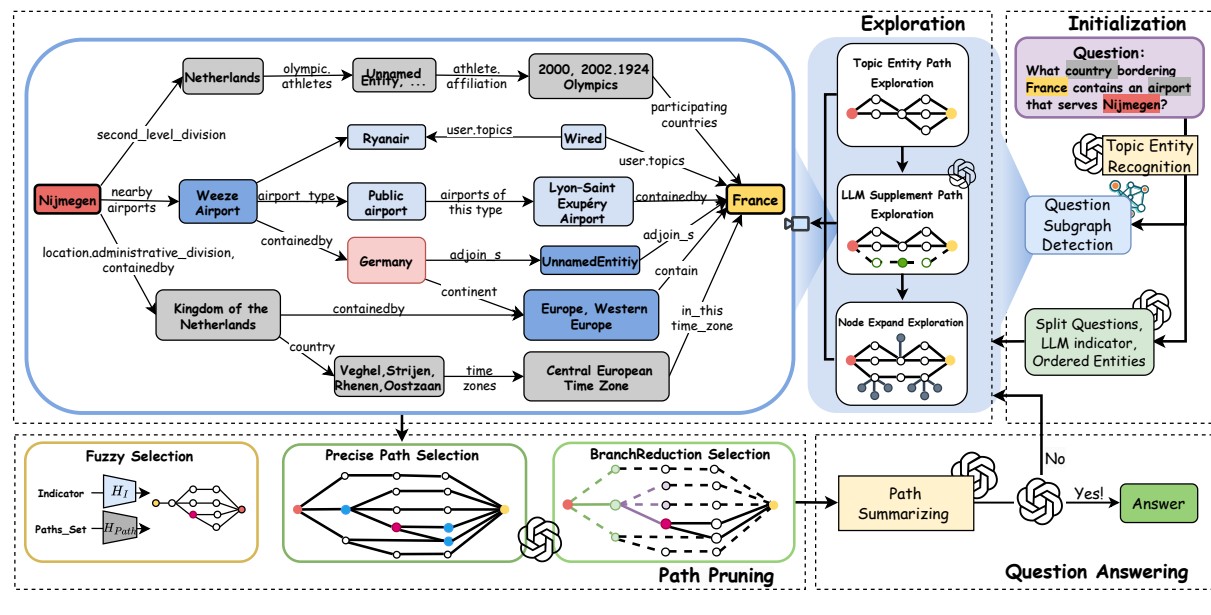

**Figure 2: Overview architecture of our proposed PoG**

the LLM to analyze the question and generate an indicator that serves as a strategy for the answer formulation process and predicting the exploration depth $D_{\text{predict}}$.

- **Exploration.** After initialization, the model retrieves topic entity paths from $\mathcal{G}_q$ through three exploration phases: topic entity path exploration, LLM supplement path exploration, and node expand exploration. All reasoning paths are constrained within the depth range $D \in [D_{\text{predict}}, D_{\text{max}}]$.
- **Path Pruning.** Following each exploration phase, PoG employs a pre-trained LM, LLM prompting, and graph structural analysis to perform a three-step beam search. The pruned paths are then evaluated in the question answering.
- **Question Answering.** Finally, LLM is prompted to assess if the pruned reasoning paths sufficiently answer the question. If not, continue exploration with deeper paths incrementally until the $D_{\text{max}}$ is exceeded or proceed to the next exploration phase.

## 4.1 Initialization

The initialization has two main stages, i.e., question subgraph detection and question analysis. The framework is shown in Figure 3.

**Question subgraph detection.** Given a question $q$, PoG initially identifies the question subgraph, which includes all the topic entities of $q$ and their $D_{\text{max}}$-hop neighbors.

Topic entity recognition. To identify the relevant subgraph, PoG first employs LLMs to extract the potential topic entities from the question. Following the identification, the process applies BERT-based similarity matching to align these potential entities with entities from KG. Specifically, as shown in Figure 3, we encode both the keywords and all entities from KG into dense vector embeddings as $H_T$ and $H_{\mathcal{G}}$. We then compute a cosine similarity matrix between these embeddings to determine the matches. For each keyword, the entities with the highest similarity scores are selected to form the

set $Topic(q)$. This set serves as the foundation for constructing the question subgraph in subsequent steps.

Subgraph detection. Upon identifying the topic entities, PoG captures the induced subgraph $\mathcal{G}_q \subseteq \mathcal{G}$ by expanding around each entity $e$ in $Topic(q)$. For each entity, we retrieve knowledge triples associated with its $D_{\text{max}}$-hop neighbors, thereby incorporating query-relevant and faithful KG information into $\mathcal{G}_q$. Through this process, we update $\mathcal{E}_q$ with newly added intermediate nodes that serve as bridging pathways between the topic entities. The result subgraph, $\mathcal{G}_q$ is defined as $(\mathcal{E}_q, \mathcal{R}_q, \mathcal{T}_q)$, where $\mathcal{E}_q$ encompasses $Topic(q)$ together with the set $\{N_{\mathcal{G}}(e, D_{\text{max}}) \mid e \in Topic(q)\}$, effectively linking all relevant entities and their connective paths within the defined hop distance. To interact with KG, we utilize the pre-defined SPARQL queries as detailed in Appendix D.

Graph pruning. To efficiently manage information overhead and reduce computational cost, we implement graph pruning on the question subgraph $\mathcal{G}_q$ using node and relation clustering alongside graph reduction techniques. As illustrated in Figure 3, node and relation clustering is achieved by compressing multiple nodes and their relations into supernodes, which aggregate information from the original entities and connections. For graph reduction, we employ bidirectional BFS to identify all paths connecting the topic entities. Based on these paths, we regenerate induced subgraphs that involve only the relevant connections, effectively excluding nodes and relations that lack strong relevance to the topic entities.

**Question Analysis.** To reduce hallucinations in LLMs, the question analysis phase is divided into two parts and executed within a single LLM call using an example-based prompt (shown in Appendix E). First, the complex question $q$ is decomposed into simpler questions based on the identified topic entities, each addressing their relationship to the potential answer. Addressing these simpler questions collectively guides the LLM to better answer the original query, thereby reducing hallucinations. Second, a LLM indicator is

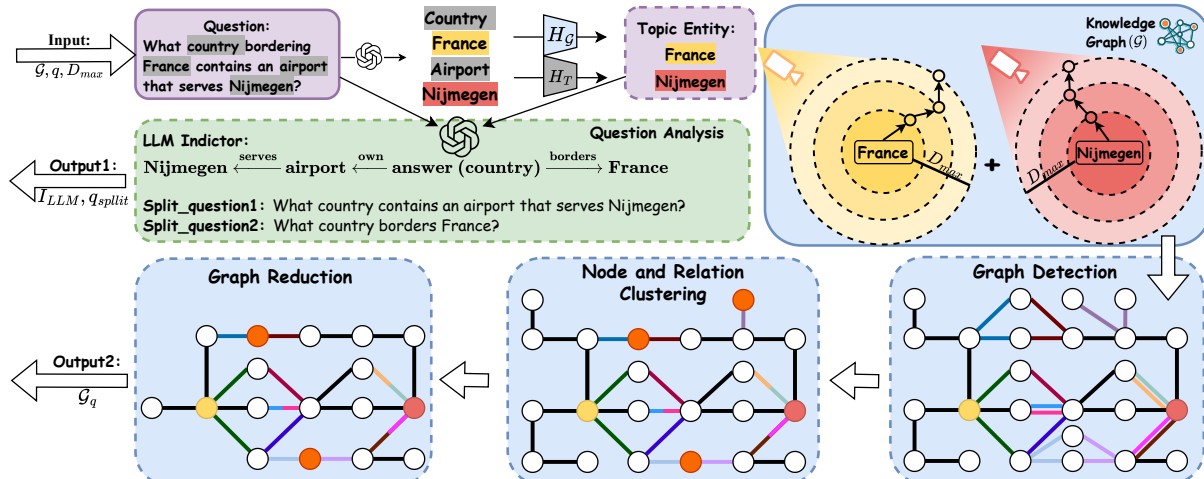

**Figure 3: Illustration of initialization**

generated, encapsulating all topic entities and predicting the answer position within a single chain of thought derived from the original question. This indicator highlights the relationships and sequence among the entities and answer. Based on this, a predicted depth $D_{\text{predict}}$ is calculated, defined as the maximum distance between the predicted answer and each topic entity. An example of question analysis is shown in Figure 3 with predicted depth 2.

## 4.2 Exploration

As discussed in Section 1, identifying reasoning paths that encompass all topic entities is essential to derive accurate answers. These paths serve as interpretable chains of thought, providing both the answer and the inference steps leading to it, a feature we refer as **interpretability**. To optimize the discovery of such paths efficiently and accurately, the exploration process is divided into three phases: topic entity path exploration, LLM supplement path exploration, and node expand exploration. After each phase, we perform path pruning and question answering. If a sufficient path is found, the process terminates; otherwise, it advances to the next phase to explore additional paths. Due to the space limitation, the pseudo-code of exploration is shown in Appendix A.1.

**Topic entity path exploration**. To reduce LLM usage and search space, PoG begins exploration from a predicted depth $D_{\text{predict}}$ rather than the maximum depth. Using the question subgraph $\mathcal{G}_q$, topic entities $Topic(q)$, LLM indicator $I_{\text{LLM}}$, and $D_{\text{predict}}$, PoG identifies reasoning paths containing all topic entities by iteratively adjusting the exploration depth $D$. Entities in $Topic(q)$ are ordered according to $I_{\text{LLM}}$ to facilitate reasoning effectively. Starting from the predicted depth $D = D_{\text{predict}}$, we employ a bidirectional BFS to derive all potential entity paths, which is defined as:

$$Paths_t = \{p \mid |Topic(q)| \times (D-1) < length(p) \leq |Topic(q)| \times D\},$$

where $p = Path_{\mathcal{G}_q}(Topic(q))$. To reduce the complexity, a pruning strategy is employed and selects the top-$W_{\text{max}}$ paths based on $Paths_t$, $I_{\text{LLM}}$, and split questions from Section 4.1. These paths are evaluated for sufficiency verification. If inadequate, $D$ is incremented until $D_{\text{max}}$ is reached. Then the next phase commences.

**LLM supplement path exploration**. Traditional KG-based LLMs reasoning often rephrase facts without utilizing the LLM's inherent knowledge. To overcome this, PoG prompts the LLM to generate predictions based on path understanding and its implicit knowledge, providing additional relevant insights. It involves generating new LLM thinking indicators $I_{\text{Sup}}$ for predicted entities $e \in Predict(q)$, aligning them with $\mathcal{E}_q$ and using text similarity to form and reorder the supplementary entity list $List_S(e) = Topic(q) + e$. Supplementary paths $Paths_s$ are then generated with a fixed depth $D_{\text{max}}$:

$$Paths_s = \{p \mid \text{length}(p) \leq |Topic(q)| \times D_{\text{max}}\},$$

where $p = Path_{\mathcal{G}_q}(List_S(e))$. These paths with new indicators are evaluated similarly to the topic entity path exploration phase. The prompting temple is shown in Appendix E.

**Node expand exploration**. If previous phases cannot yield sufficient paths, PoG proceeds to node expansion. Unlike previous methods [26, 34] that separately explore relations and entities, PoG explores both simultaneously, leveraging clearer semantic information for easier integration with existing paths. During the exploration, PoG expands unvisited entities by 1-hop neighbors in $\mathcal{G}$. New triples are merged into existing paths to form the new paths, followed by pruning and evaluation.

## 4.3 Path Pruning

As introduced in Section 2, KGs contain vast amounts of facts, making it impractical to involve all relevant triples in the LLM's context due to high costs. To address this complexity and reduce LLM overhead, we utilize a three-step beam search for path pruning. The corresponding pseudo-code can be found in Appendix A.2.

**Fuzzy selection**. Considering that only a small subset of the generated paths is relevant, the initial step of our beam search involves fuzzy selection by integrating a pre-trained language model (e.g. SentenceBERT [32]), to filter the irrelevant paths quickly. As shown in Figure 2, we encode the LLM indicator $I_{\text{LLM}}$ (or $I_{\text{Sup}}$) and all reasoning paths into vector embeddings, denoted as $H_I$ and $H_{Paths}$, and calculate cosine similarities between them. The top-$W_1$ paths with the highest similarity scores are selected for further evaluation.

**Precise path selection**. Following the initial fuzzy selection, the number of candidate paths is reduced to $W_1$. At this stage, we prompt the LLM to select the top-$W_{max}$ reasoning paths most likely to contain the correct answer. The specific prompt used to guide LLM in selection phase can be found in Appendix E.

**Branch reduced selection**. Considering that paths are often represented in natural language and can be extensive, leading to high processing costs for LLMs, we implement a branch reduced selection method integrated with the graph structure. This method effectively balances efficiency and accuracy by further refining path selection. Starting with $D = 1$, for each entity $e$ in the entity list, we extract the initial $D$-step paths from every path in the candidate set $Paths_c$ into a new set $Paths_e$. If the number of $Paths_e$ exceeds the maximum designated width $W_{max}$, these paths are pruned using precise path selection. The process iterates until the number of paths in $Paths_c$ reaches $D_{max}$. For example, as illustrated in Figure 2, with $W_{max} = 1$, only the initial step paths (depicted in green) are extracted for further examination, while paths represented by dashed lines are pruned. This selection method enables efficient iterative selection by limiting the number of tokens and ensuring the relevance and conciseness of the reasoning paths.

**Beam search strategy**. Based on the three path pruning methods above, PoG can support various beam search strategies, ranging from non-reliant to fully reliant on LLMs. These strategies are selectable in a user-friendly manner, allowing flexibility based on the specific requirements of the task. We have defined four such strategies in Algorithm 2 of Appendix A.2.

### 4.4 Question Answering

Based on the pruned paths in Section 4.3, we introduce a two-step question-answering method.

**Path Summarizing**. To address hallucinations caused by paths with excessive or incorrect text, we develop a summarization strategy by prompting LLM to review and extract relevant triples from provided paths, creating a concise and focused path. Details of the prompts used are in Appendix E.

**Question answering**. Based on the current reasoning path derived from path pruning and summarizing, we prompt the LLM to first evaluate whether the paths are sufficient for answering the split question and then the main question. If the evaluation is positive, LLM is prompted to generate the answer using these paths, along with the question and question analysis results as inputs, as shown in Figures 2. The prompts for evaluation and generation are detailed in Appendix E. If the evaluation is negative, the exploration process is repeated until completion. If node expand exploration reaches its depth limit without yielding a satisfactory answer, LLM will leverage both provided and inherent knowledge to formulate a response. Additional details on the prompts can be found in Appendix E.

## 5 EXPERIMENTS

**Experimental settings** We evaluate PoG on five KGQA datasets, i.e., CWQ [35], WebQSP [45], GrailQA [12], SimpleQuestions [29], and WebQuestions [3]. PoG is tested against methods without external knowledge (IO, CoT[40], SC[39]) and the state-of-the-art (SOTA) approaches with external knowledge, including prompting-based and fine-tuning-based methods. Freebase [5] serves as the

background knowledge graph for all datasets. Experiments are conducted using two LLMs, i.e., GPT-3.5 (GPT-3.5-Turbo) and GPT-4. Following prior studies, we use exact match accuracy (Hits@1) as the evaluation metric. Due to the space limitation, detailed experimental settings, including dataset statistics, baselines, and implementation details, are provided in Appendix C.

**PoG setting**. We adopt the `Fuzzy + Precise Path Selection` strategy in Algorithm 2 of Appendix A.2 for PoG, with $W_1 = 80$ for fuzzy selection. Additionally, we introduce **PoG-E**, which randomly selects one relation from each edge in the clustered question subgraph to evaluate the impact of graph structure on KG-based LLM reasoning. $W_{max}$ and $D_{max}$ are 3 by default for beam search.

### 5.1 Main Results

Since PoG leverages external knowledge to enhance LLM reasoning, we first compare it with other methods that utilize external knowledge. Although PoG is a training-free, prompting-based method and has natural disadvantages compared to fine-tuned methods trained on evaluation data. As shown in Table 1, PoG with GPT-3.5-Turbo still achieves new SOTA performance across most datasets. Additionally, PoG with GPT-4 surpasses fine-tuned SOTA across all the multi-hop and open-domain datasets by an average of 17.3% and up to 28.3% on the WebQuestions dataset. Comparing all the in-context learning (ICL) methods, PoG with GPT-3.5-Turbo surpasses all the previous SOTA methods. When comparing PoG with GPT-3.5-Turbo against SOTA using GPT-4, PoG outperforms the SOTA by an average of 12.9% and up to 23.9%. When using the same LLM, PoG demonstrates substantial improvements: with GPT-3.5-Turbo, it outperforms SOTA by an average of 21.2% and up to 27.3% on the WebQuestions dataset; with GPT-4, it outperforms SOTA by 16.6% on average and up to 26.7% on the WebQuestions dataset. Additionally, PoG with GPT-3.5-Turbo outperforms methods without external knowledge (e.g., IO, CoT, SC prompting) by 62% on GrailQA and 60.5% on Simple Questions. These results show that incorporating external knowledge graphs significantly enhances reasoning tasks. PoG-E also achieves excellent results. Under GPT-4, PoG-E surpasses all SOTA in ICL by 14.1% on average and up to 24.1% on the WebQuestions dataset. These findings demonstrate that the graph structure is crucial for reasoning tasks, particularly for complex logical reasoning. By integrating the structural information of the question within the graph, PoG enhances the deep reasoning capabilities of LLMs, leading to superior performance.

### 5.2 Ablation Study

We perform various ablation studies to understand the importance of different factors in PoG. These ablation studies are performed with GPT-3.5-Turbo on two subsets of the CWQ and WebQSP test sets, each containing 500 randomly sampled questions.

**Does search depth matter?** As described, PoG's dynamic deep search is limited by $D_{max}$. To assess the impact of $D_{max}$ on performance, we conduct experiments with depth from 1 to 4. The results, shown in Figures 4(a) and (c), indicate that performance improves with increased depth, but the benefits diminish beyond a depth of 3. Figures 4(b) and (d), showing which exploration phase the answer is generated from, reveal that higher depths reduce the effectiveness of both LLM-based path supplementation and node

**Table 1: Results of PoG across various datasets, compared with the state-of-the-art (SOTA) in Supervised Learning (SL) and In-Context Learning (ICL) methods. The highest scores for ICL methods are highlighted in bold, while the second-best results are underlined. The Prior FT (Fine-tuned) SOTA includes the best-known results achieved through supervised learning.**

| Method | Class | LLM | Multi-Hop KGQA | | | Single-Hop KGQA | Open-Domain QA |
|--------|-------|-----|------|---------|--------|-----------------|----------------|
| | | | CWQ | WebQSP | GrailQA | Simple Questions | WebQuestions |
| *Without external knowledge* | | | | | | | |
| IO prompt | - | GPT-3.5-Turbo | 37.6 | 63.3 | 29.4 | 20.0 | 48.7 |
| CoT | - | GPT-3.5-Turbo | 38.8 | 62.2 | 28.1 | 20.3 | 48.5 |
| SC | - | GPT-3.5-Turbo | 45.4 | 61.1 | 29.6 | 18.9 | 50.3 |
| *With external knowledge* | | | | | | | |
| Prior FT SOTA | SL | - | 70.4[9] | 85.7[25] | 75.4[11] | 85.8[1] | 56.3[18] |
| KB-BINDER[22] | ICL | Codex | - | 74.4 | 58.5 | - | - |
| ToG/ToG-R[34] | ICL | GPT-3.5-Turbo | 58.9 | 76.2 | 68.7 | 53.6 | 54.5 |
| ToG-2.0[26] | ICL | GPT-3.5-Turbo | - | 81.1 | - | - | - |
| ToG/ToG-R[34] | ICL | GPT-4 | 69.5 | 82.6 | 81.4 | 66.7 | 57.9 |
| PoG-E | ICL | GPT-3.5-Turbo | 71.9 | 90.9 | 87.6 | 78.3 | 76.9 |
| PoG | ICL | GPT-3.5-Turbo | 74.7 | 93.9 | 91.6 | 80.8 | 81.8 |
| PoG-E | ICL | GPT-4 | 78.5 | 95.4 | 91.4 | 81.2 | 82.0 |
| PoG | ICL | GPT-4 | **81.4** | **96.7** | **94.4** | **84.0** | **84.6** |

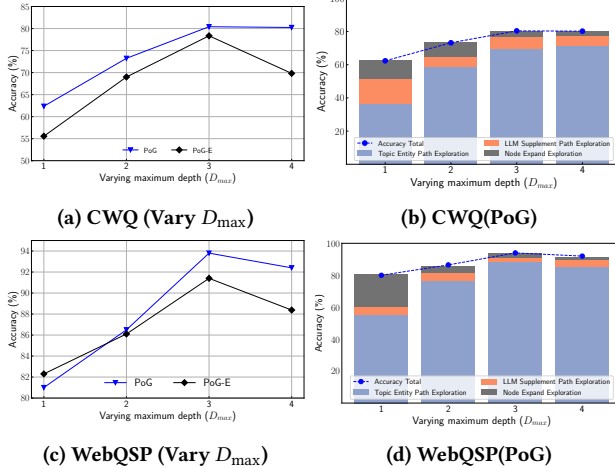

**(a) CWQ (Vary $D_{max}$)**   **(b) CWQ(PoG)**

**(c) WebQSP (Vary $D_{max}$)**  **(d) WebQSP(PoG)**

**Figure 4: The accuracy of PoG and PoG-E among CWQ and WebQSP datasets by varying different $D_{max}$.**

exploration. Excessive depth leads to LLM hallucinations and difficulties in managing long reasoning paths. Therefore, we set the maximum depth to 3 for experiments to balance performance and computational efficiency. Additionally, even at lower depths, PoG maintains strong performance by effectively combining the LLM's inherent knowledge with the structured information from the KG.
**Compare the effect of different beam searches**. As introduced in Section 4.3, PoG supports various beam search strategies, ranging from non-reliant to fully reliant on LLMs, selectable in a user-friendly manner. To evaluate the computational cost and performance, we test four cases outlined in Algorithm 2. In the 3-Step

Beam Search case, we set $W_2 = 20$ for internal narrowing. The Fuzzy Selection approach, as described in Section 4.3, utilizes all candidate paths and a LLM-generated indicator for encoding and comparison. We report accuracy, average LLM calls in total, and average token input during the path pruning for each beam search strategy applied to PoG in Table 2. The experimental results for PoG-E are provided in Table 6 in Appendix B.1. These results indicate that PoG with Fuzzy and Precise Path Selection achieves the highest accuracy. Additionally, the BranchReduced Selection method, which leverages the graph structure, not only delivers excellent results but also reduces token usage by over 50% with only a ±2% difference in accuracy compared to the best-performing strategy. Furthermore, the Fuzzy Selection method, which employs lightweight models instead of relying solely on LLMs, also demonstrates strong performance. These results validate the effectiveness of our beam search strategies and underscore the importance of structure-based faithful path reasoning.

## 5.3 Effectiveness Evaluation

**Effective evaluation on multi-entity questions**. To evaluate PoG's performance on multi-entity questions, we report the accuracy on all test sets by categorizing questions based on the number of topic entities. The results, shown in Table 3, demonstrate that, despite the increased complexity of multi-entity questions compared to single-entity ones, PoG maintains excellent accuracy, achieving up to 93.9% on the WebQSP dataset. This underscores the effectiveness of our structure-based model in handling complex multi-entity queries. Notably, the slightly lower performance on the GrailQA dataset can be attributed to some questions lacking matched topic entities, which prevents effective reasoning using KG.

**Table 2: Performance comparison of PoG with different beam search methods on CWQ and WebQSP.**

| PoG | Evaluation | CWQ | WebQSP |
|---|---|---|---|
| w/ Fuzzy Selection | Accuracy | 57.1 | 86.4 |
| | Token Input | - | - |
| | LLM Calls | 6.8 | 6.5 |
| w/ Fuzzy and | Accuracy | 79.3 | 93.0 |
| BranchReduced Selection | Token Input | 101,455 | 328,742 |
| | LLM Calls | 9.7 | 9.3 |
| w/ Fuzzy and | Accuracy | **81.4** | **93.9** |
| Precise Path Selection | Token Input | 216,884 | 617,448 |
| | LLM Calls | 9.1 | 7.5 |
| w/ 3-Steps Beam Search | Accuracy | 79.8 | 91.9 |
| | Token Input | 102,036 | 369,175 |
| | LLM Calls | 8.8 | 9.0 |

**Table 3: Performance of PoG and PoG-E on multi-entity and single-entity questions of all datasets. The symbol '-' indicates no multi-entity question inside.**

| Question Set | CWQ | WebQSP | GrailQA | WebQuestions | Simple Questions |
|---|---|---|---|---|---|
| **PoG** with GPT-3.5-Turbo | | | | | |
| Single-entity | 70.3 | 93.9 | 92.1 | 81.7 | 78.3 |
| Multi-entity | 80.2 | 93.1 | 70.7 | 82.8 | - |
| **PoG-E** with GPT-3.5-Turbo | | | | | |
| Single-entity | 67.5 | 91 | 88.2 | 76.8 | 80.8 |
| Multi-entity | 77.5 | 82.8 | 76.0 | 82.8 | - |

**Effective evaluation on multi-hop reasoning**. To assess PoG's performance on multi-hop reasoning tasks, we analyze accuracy by categorizing questions based on the length of their ground-truth SPARQL queries. We randomly sample 1,000 questions from CWQ and WebQSP datasets and determine the reasoning length of each question by counting the number of relations in their ground-truth SPARQL queries. The distribution of questions with varying reasoning lengths is illustrated in Figure 5. We evaluate the performance of PoG and PoG-E across different ground-truth lengths to understand their effectiveness under varying query complexities. As shown in Figure 6, the performance of PoG and PoG-E remains consistent across different reasoning lengths. Even at the highest length levels in the WebQSP dataset, PoG achieves excellent accuracy, reaching up to 90%. Notably, although some questions have ground-truth lengths of eight or more, PoG successfully addresses them without matching the ground-truth length, demonstrating its ability to explore novel paths by effectively combining the LLM's inherent knowledge with the structured information from the KG. These results demonstrate the effectiveness of PoG in handling complex multi-hop reasoning tasks.

**Graph structure pruning**. To evaluate the effectiveness of the graph pruning method proposed in Section 4.1, we conduct experiments using 200 random samples from each dataset. We report the average number of entities per question before and after graph reduction, as well as the proportion of entities reduced, in Table 4. The results indicate that up to 75% of entities in the WebQSP

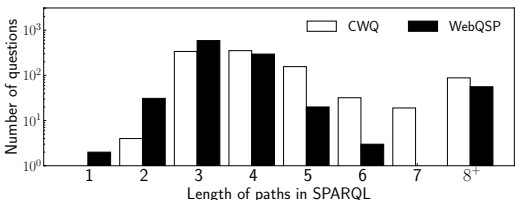

**Figure 5: The lengths of the ground-truth SPARQL queries within the CWQ and WebQSP datasets.**

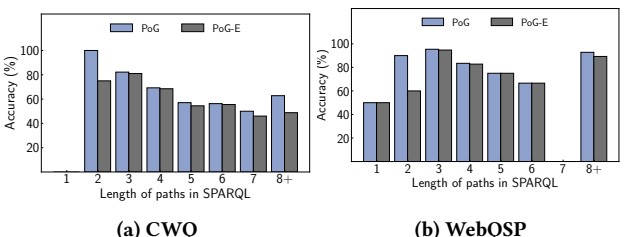

(a) CWQ          (b) WebQSP

**Figure 6: The accuracy of PoG and PoG-E on the CWQ and WebQSP datasets, categorized by the different lengths of the ground-truth answers for each question.**

**Table 4: The illustration of graph size reduction.**

| | CWQ | WebQSP | GrailQA | WebQuestions |
|---|---|---|---|---|
| Ave Entity Number | 3,540,267 | 243,826 | 62,524 | 240,863 |
| Ave Entity Number After Pruned | 1,621,055 | 182,673 | 30,267 | 177,822 |
| Ave Entitiy Reduction Proportion (%) | **46%** | **75%** | **48%** | **74%** |

dataset can be pruned before path exploration. This demonstrates the effectiveness of eliminating irrelevant data from the outset.

**Case study: interpretable reasoning**. We also conduct the case study to demonstrate interpretability of PoG, we present three reasoning examples in Table 8 of Appendix B.5. These examples feature questions with one, two, and three entities, respectively. Through the case study, we showcase PoG's effectiveness in handling multi-entity and multi-hop tasks by providing faithful and interpretable reasoning paths that lead to accurate answers.

To further evaluate the effectiveness and efficiency of PoG, we perform additional experiments, including prompt setting ablation (Appendix B.1), reasoning faithfulness analysis (Appendix B.2), error analysis (Appendix B.3), LLM cost analysis (Appendix B.4), and graph reduction and path pruning case study (Appendix B.5).

## 6 CONCLUSION

In this paper, we introduce Paths-over-Graphs (PoG), a novel method that integrates LLMs with KGs to enable faithful and interpretable reasoning. PoG addresses complex reasoning tasks through a three-phase dynamic multi-hop path exploration, combining the inherent knowledge of LLMs with factual information from KGs. Efficiency is enhanced by graph-structured pruning and a three-step pruning process to effectively narrow down candidate paths. Extensive experiments on five public datasets demonstrate that PoG outperforms existing baselines, showcasing its superior reasoning capabilities and interoperability.

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

# A ALGORITHM

## A.1 Exploration

We summarize the comprehensive algorithmic procedure for exploration detailed in Section 4.2 as presented in Algorithm 1.

---

**Algorithm 1**: Exploration

---

**Input** : Question subgraph ($\mathcal{G}_q$), source KG ($\mathcal{G}$), question and split question ($Q = q + q_{split}$), topic entities ($Topic(q)$), LLM indicator ($I_{LLM}$), predict depth ($D_{predict}$), maximum depth ($D_{max}$), maximum width ($W_{max}$), and path pruning case ($case$)

**Output** : PoG answers ($a(q)$), final reasoning path ($Paths_f(q)$)

       /* **Start with topic entity path exploration** */
1   $List_T \leftarrow$ Reorder($Topic(q), I_{LLM}$), $D \leftarrow \min(D_{predict}, D_{max})$;
2   **while** $D \leq D_{max}$ **do**
3     $Paths_t \leftarrow$ EntityPathFind ($List_T, D, \mathcal{G}_q$);
4     PathPruning($Paths_t, Q, I_{LLM}, W_{max}, D_{max}, List_T, case$);
5     $Answer, Paths_T \leftarrow$ QuestionAnswering($Paths_t, Q, I_{LLM}$);
6     **if** "{Yes}" in $Answer$ **then return** $Answer, Paths_T$;
7     **else** $D \leftarrow D + 1$;

       /* **LLM supplement path exploration procedure** */
8   $Paths_s \leftarrow [\,]$;
9   $Predict(q) \leftarrow$ SupplementPrediction($Paths_T, Q, I_{LLM}$);
10 **for each** $e, I_{sup(e)} \in Predict(q)$ **do**
11     $List_S \leftarrow$ Reorder ($List_T + e, I_{sup(e)}$);
12     $Paths'_s \leftarrow$ EntityPathFind ($List_S, D_{max}, \mathcal{G}_q$);
13     $Paths_s \leftarrow Paths_s +$ FuzzySelect ($Paths'_s, I_{sup(e)}, W_{max}$);
14 PathPruning($Paths_s, Q, I_{LLM}, W_{max}, D_{max}, List_S, case$);
15 $Answer, Paths_S \leftarrow$ QuestionAnswering($Paths_s, Q, I_{LLM}$);
16 **if** "{Yes}" in $Answer$ **then return** $Answer, Paths_S$ ;

       /* **Node expand exploration procedure** */
17 $Visted \leftarrow \emptyset, D \leftarrow 1, Paths_e \leftarrow Paths_T + Paths_S$;
18 PathPruning($Paths_e, Q, I_{LLM}, W_{max}, D_{max}, List_T, case$);;
19 **while** $D \leq D_{max}$ **do**
20     **for each** $e \in ExtractEntity(Paths_e) \wedge e \notin Visted$ **do**
21       $Related\_edges =$ Find_1_hop_Edges($\mathcal{G}, e$);
22       $Paths_e \leftarrow$ MergeTogether($Paths_e, Related\_edge$);
23     PathPruning($Paths_e, Q, I_{LLM}, W_{max}, D_{max}, List_T, case$);
24     $Answer, Paths_e \leftarrow$ QuestionAnswering($Paths_e, Q, I_{LLM}$);
25     **if** "{Yes}" in $Answer$ **then return** $Answer, Paths_e$;
26     **else** $Visted \leftarrow Visted \cup e; D \leftarrow D + 1$;
27 $Paths_l \leftarrow Paths_T + Paths_S + Paths_E$ ;
28 PathPruning($Paths_l, Q, I_{LLM}, W_{max}, D_{max}, List_T, case$);
29 $Answer, Paths_L \leftarrow$ QuestionAnsweringFinal($Paths_l, Q, I_{LLM}$);
30 **Return** $Answer, Paths_L$;

---

## A.2 Path Pruning

We summarize the comprehensive algorithmic procedure of path pruning detailed in Section 4.3 as presented in Algorithm 2.

---

**Algorithm 2**: PathPruning

---

**Input** : Candidate paths($Paths_c$), question and split question ($Q = q + q_{split}$), indicator ($I$), maximum width ($W_{max}$), maximum depth ($D_{max}$), entity list ($list$)

**Output** : Pruned candidate paths ($Paths_c$)

1   **if** $Case =$ Fuzzy Selection Only **then**
2     FuzzySelect($Paths_c, Q, I, W_{max}$);
3   **else if** $Case =$ Fuzzy + Precise Path Selection **then**
4     FuzzySelect($Paths_c, Q, I, W_1$);
5     FullPathSelect($Paths_c, Q, I, W_{max}$);
6   **else if** $Case =$ Fuzzy + Branch Reduced Selection **then**
7     FuzzySelect($Paths_c, Q, I, W_1$);
8     BranchReduceSelect($Paths_c, Q, I, W_{max}, D_{max}, list$);
9   **else if** $Case =$ Fuzzy + Branch Reduced + Precise Path **then**
       /* case = 3-Step Beam Search */
    FuzzySelect($Paths_c, Q, I, W_1$);
10     BranchReduceSelect($Paths_c, Q, I, W_2, D_{max}, list$);
11     FullPathSelect($Paths_c, Q, I, W_{max}$);

12 **Procedure** BranchReduceSelect($Paths_c, Q, I, W, D_{max}, list$)
13 $D \leftarrow 1, Paths_e \leftarrow \emptyset$;
14 **while** $|Paths_c| \geq W \wedge D \leq D_{max}$ **do**
15     **for each** $e \in list$ **do**
16       $Paths_e \leftarrow Paths_e \cup$ ExtractHeadSteps($Paths_c, e, D$);
17     **if** $|Paths_e| > W$ **then**
18       FullPathSelect($Paths_e, Q, I, W$);
19       $Paths_c \leftarrow$ IntersectMatchUpdate($Paths_e, Paths_c$);
20       $Paths_e \leftarrow \emptyset$;
21     $D \leftarrow D + 1$;
22 **if** $|Paths_c| > W$ **then** FullPathSelect($Paths_c, Q, I, W$);

---

# B  EXPERIMENT

## B.1  Additional Ablation Study

**How do summary prompts affect?** Inspired by GoT [4], we utilize summary prompts to reduce LLM hallucinations and decrease computational costs. To evaluate their impact, we conduct an ablation study comparing PoG and PoG-E with and without path summarization. We measure both accuracy and average token input to the LLM API during the path pruning phase to measure efficiency and effectiveness. The results, present in Tabel 5, show that using graph summaries increases accuracy by up to 10% on the CWQ dataset with PoG-E, while reducing token input by up to 36% on WebQSP. These results indicate hat path summarization effectively minimizes LLM hallucinations, enhances the LLM's understanding of the explored paths, facilitates answer retrieval, enables earlier termination of the reasoning process, and reduces costs.

**Table 5: Performance comparison of PoG and PoG-E with and without path summarizing on CWQ and WebQSP datasets.**

| Method | Evaluation | CWQ | WebQSP |
| --- | --- | --- | --- |
| **PoG** | | | |
| w/ Path Summarizing | Accuracy | **81.4** | **93.9** |
| | Token Input | 216,884 | 297,359 |
| w/o Path Summarizing | Accuracy | 74.7 | 91.9 |
| | Token Input | 273,447 | 458,545 |
| **PoG-E** | | | |
| w/ Path Summarizing | Accuracy | **80.4** | **91.4** |
| | Token Input | 314,747 | 273,407 |
| w/o Path Summarizing | Accuracy | 70.4 | 90.4 |
| | Token Input | 419,679 | 428,545 |

**How do different beam searches affect PoG-E?** As detailed in Section 5.2, Table 6 presents the accuracy, average LLM calls in total, and average token input during the path pruning for each beam search strategy applied to PoG-E. PoG-E using `Fuzzy and Precise Path Selection` achieves the highest accuracy. The `BranchReduced Selection` method, which leverages graph structure, not only delivers excellent results but also reduces token usage by up to 65% with only a ±4.3% accuracy drop compared to the best strategy. Additionally, the `Fuzzy Selection` method, employing lightweight models instead of solely relying on LLMs, also demonstrates strong performance.

## B.2  Reasoning Faithfulness Analysis

**Overlap Ratio between Explored Paths and Ground-Truth Paths**. We analyzed correctly answered samples from three datasets to investigate the overlap ratio between the paths $P$ explored by PoG and the ground-truth paths $S$ in SPARQL queries. The overlap ratio is defined as the proportion of overlapping relations to the total number of relations in the ground-truth SPARQL path:

$$Ratio(P) = \frac{|Relation(P) \cap Relation(S)|}{|Relation(S)|},$$

where $Relation(P)$ denotes the set of relations in path $P$. Figure 7 illustrates the distribution of questions across different overlap ratios. For the WebQSP dataset, PoG achieves the highest proportion

**Table 6: Performance comparison of PoG-E with different beam search Methods among CWQ and WebQSP datasets.**

| PoG-E | Evaluation | CWQ | WebQSP |
| --- | --- | --- | --- |
| w/ FuzzySelect | Accuracy | 62.31 | 82.3 |
| | Token Input | - | - |
| | Ave LLM Calls | 6 | 6.3 |
| w/ Fuzzy and BranchReduced Selection | Accuracy | 71.9 | 88.4 |
| | Token Input | 128,407 | 371,083 |
| | Ave LLM Calls | 9.4 | 9.1 |
| w/ Fuzzy and Precise Path Selection | Accuracy | **80.4** | **91.4** |
| | Token Input | 344,747 | 603,261 |
| | Ave LLM Calls | 8.3 | 7.4 |
| w/ 3-Steps Beam Search | Accuracy | 73.87 | 89.4 |
| | Token Input | 120,159 | 411,283 |
| | Ave LLM Calls | 8.3 | 9.1 |

of fully overlapping paths with the ground truth, reaching approximately 60% accuracy. In contrast, PoG-E applied to the GrailQA dataset shows the highest proportion of paths with up to 70% non-overlapping relations, indicating that PoG-E explores novel paths to derive the answers. The different results between PoG and PoG-E are due to PoG-E's strategy of randomly selecting one related edge from each clustered edge. This approach highlights the effectiveness of our structure-based path exploration method in generating diverse and accurate reasoning paths.

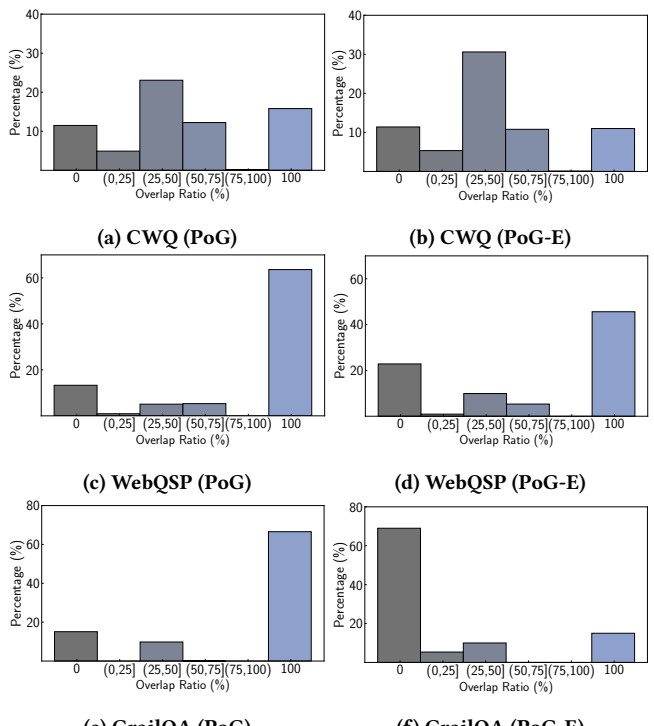

(a) CWQ (PoG)  (b) CWQ (PoG-E)

(c) WebQSP (PoG)  (d) WebQSP (PoG-E)

(e) GrailQA (PoG)  (f) GrailQA (PoG-E)

**Figure 7: The path overlap ratio of PoG and PoG-E among CWQ, WebQSP, and GrailQA datasets.**

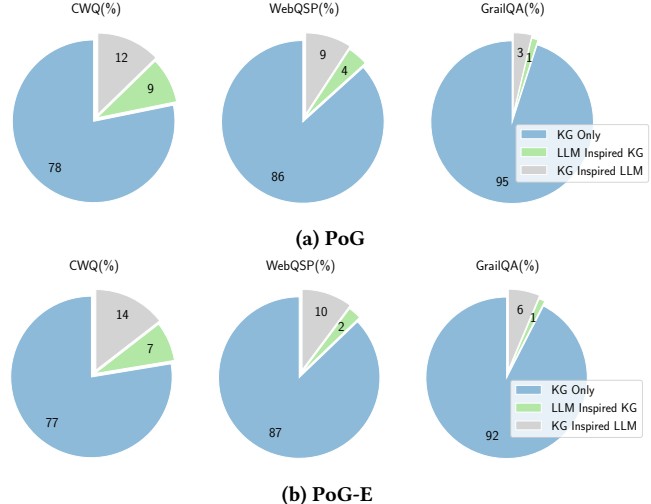

(a) PoG

(b) PoG-E

**Figure 8: The proportions of answer evidence of PoG and PoG-E among CWQ, WebQSP, and GrailQA datasets.**

**Evidence of answer exploration sources**. We conduct an analysis of correctly answered samples from three datasets to investigate the sources of evidence used by the LLM in generating answers, as illustrated in Figure 8. Specifically, we categorize all generated answers into three cases: KG only, LLM-inspired KG, and KG-inspired LLM. In the *KG only* scenario, answers are generated solely based on KG paths. The *LLM-inspired KG* case involves the LLM predicting an answer using its inherent knowledge and subsequently using the KG to verify its correctness. Conversely, in the *KG-inspired LLM* case, the paths generated by the KG are insufficient to reach the answer, and the LLM supplements the reasoning using its inherent knowledge. As shown in the figure, up to 14% of answers are generated through the KG-inspired LLM approach, and up to 9% involve LLM-inspired KG path supplementation. Compared to previous work that integrates LLM inherent knowledge with KG data[34], PoG more effectively leverages the strengths of both sources. These results demonstrate that PoG is a faithful reasoning method that primarily relies on KG-based reasoning while being supplemented by the LLM, ensuring both accuracy and interpretability in answer generation.

## B.3 Error Analysis

To further analyze the integration of LLMs and KGs, we conduct an error analysis on the CWQ, WebQSP, and GrailQA datasets. We categoriz errors into four types: (1) answer generation error, (2) refuse error, (3) format error, and (4) other hallucination errors. Note that answer generation error occurs when PoG provides an accurate reasoning path, but the LLM fails to extract the correct answer from it. The distribution of these error types is illustrated in Figure 9. The results indicate that using more powerful LLMs reduces the number of "other hallucination errors," "refuse errors," and "answer generation errors," as the model offers enhanced reasoning capabilities based on the retrieved data. Specifically, the reduction in "answer generation errors" shows the reasoning paths provided by PoG are effectively utilized by more advanced LLMs. However, we

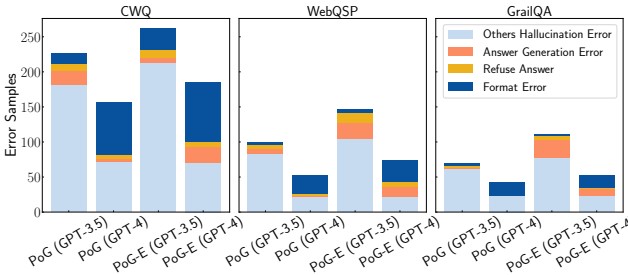

**Figure 9: The error instances and categories of PoG and PoG-E in the CWQ, WebQSP, and GrailQA datasets.**

observe an increase in "format errors" with more powerful LLMs, which may be attributed to their greater creative flexibility.

## B.4 LLM Calls Cost Analysis

To further evaluate the cost and efficiency of utilizing LLMs, we conducted an analysis of LLM calls on the CWQ, WebQSP, and GrailQA datasets. Initially, we examined the proportion of questions answered with varying numbers of LLM calls, as depicted in Figure 10. The results indicate that the majority of questions are answered within nine LLM calls across all datasets, with approximately 80% and 50% of questions being resolved within six calls on CWQ and WebQSP, respectively. These findings demonstrate PoG's efficiency in minimizing LLM usage costs. Furthermore, we compared the average number of LLM calls required by PoG with the current SOTA method, ToG [34], as shown in Table 7. Since we utilized identical datasets for WebQSP, GrailQA, Simple Questions, and WebQuestions, we report the ToG results from their paper. The comparison reveals that PoG achieves comparable or superior accuracy while reducing the number of LLM calls by up to 40% on the GrailQA dataset compared to ToG. This improvement is attributed to PoG's dynamic exploration strategy, which avoids starting from scratch, and its effective use of graph structures to prune irrelevant information, thereby significantly decreasing computational costs.

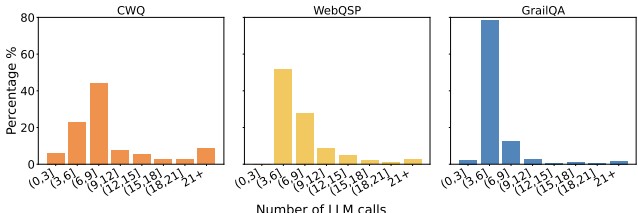

**Figure 10: The proportion of question of PoG and PoG-E by different LLM Calls among CWQ, WebQSP, and GrailQA datasets**

**Table 7: Average LLM calls per question of PoG and ToG among all datasets.**

| Method | CWQ | WebQSP | GrailQA | Simple Questions | WebQuestions |
|---|---|---|---|---|---|
| **PoG** | **10.7** | **8.3** | **6.5** | **6.1** | **9.3** |
| **ToG** | - | 11.2 | 10.6 | 8.7 | 10.5 |

## B.5 Case Study

**Case study: graph reduction and path pruning**. We conducted a case study using the example question presented in Figure 2 to illustrate the effects of graph pruning and path pruning on the graph structure. Figure 11(a) shows the results of graph pruning, where vertices in blue are selected as part of the question subgraph, and vertices in black are pruned. In this sample, the number of entities is reduced from 16,740 to 1,245, resulting in a 92% reduction of vertices. Figures 11(b) and 11(c) demonstrate the question subgraph induced by the blue vertices in Figure 11(a) and the results after applying fuzzy and precise path selection. In these figures, vertices in blue represent the selected entity after each pruning, vertices in yellow represent the topic entities, and the vertex in red denotes the final answer entity. From these graphs, we observe that utilizing the graph structure allows for the rapid pruning of irrelevant vertices, ensuring that the reasoning paths remain faithful and highly relevant to the question, since all vertices within question subgraph are interconnected with all topic entities, thereby maintaining the integrity and relevance of the reasoning process.

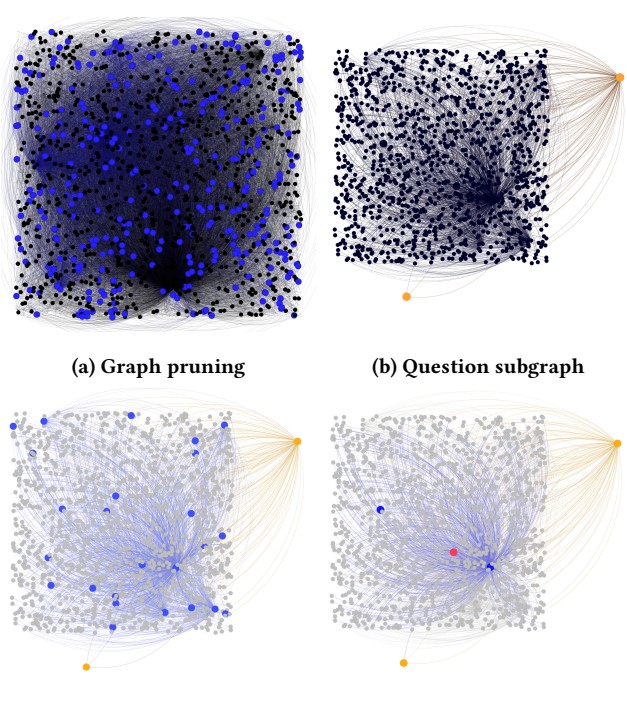

**(a) Graph pruning**   **(b) Question subgraph**

**(c) Fuzzy selection**   **(d) Precise selection**

**Figure 11: Visualization of graph reduction and Path selection.**

**Case study: interpretable reasoning**. In this section, we present Table 8, which illustrates PoG's interpretability through case studies involving questions with one, two, and three entities. These examples demonstrate PoG's effectiveness in handling multi-entity and multi-hop tasks by providing clear and understandable reasoning paths that lead to accurate answers.

## C  EXPERIMENT DETAILS

**ExDriment Datasets**. To evaluate PoG's capability in multi-hop knowledge-intensive reasoning tasks, we assess it on four KBQA datasets: three multi-hop datasets (CWQ [35], WebQSP [45], GrailQA [12]) and one single-hop dataset (SimpleQuestions [29]). Additionally, to examine PoG on more general tasks, we include an open-domain QA dataset, WebQuestions. For the evaluation of large datasets such as CWQ, GrailQA, and SimpleQuestions, we utilize a random sample of 1,000 test cases from CWQ and employ the 1,000 samples previously reported by ToG [34] to facilitate a comparison with the SOTA while also minimizing computational costs. Freebase serves as the background knowledge graph for all datasets, which encompasses approximately 88 million entities, 20,000 relations, and 126 million triples [5, 25]. The statistics of the datasets utilized in this study are detailed in Table 9.

**Baselines**. Inspired by ToG [34], we compare our method with standard prompting (IO), Chain-of-Thought (CoT), and Self-Consistency(SC) promptings with six in-context exemplars and "step-by-step" reasoning chains. For each dataset, we also include previous SOTA works for comparison. For a fair play, we compare with previous SOTA among all prompting-based methods and previous SOTA among all methods respectively. Since ToG is the current SOTA prompting-based method, we directly refer to their results and those of other baselines reported in their paper for comparisons.

**Experimental implementation**. Leveraging the plug-and-play convenience of our framework, we experiment with two LLMs: GPT-3.5 and GPT-4. We use the OpenAI API to access GPT-3.5 (GPT-3.5-turbo) and GPT-4. Aligning with ToG, we set the temperature parameter to 0.4 during the exploration process (to increase diversity) and to 0 during the reasoning process (to ensure reproducibility). The maximum token length for generation is set to 256. In all experiments, we set both $W_{max}$ and $D_{max}$ to 3 for beam search. All the experiments are conducted on a machine with Intel Xeon Gold 6248R CPU, Nvidia A5000 GPU and 512GB memory.

**Table 9: Statistics of the datasets used in this paper. [†] denotes we randomly select 1,000 samples from the CWQ test set to create the experiment testing set due to the abundance of test samples. [*] denotes that we utilize the 1,000 samples reported by ToG [34] to compare with the state-of-the-art.**

| Dataset | Answer Format | Test | Train |
|---|---|---|---|
| ComplexWebQuestions (CWQ) [†] | Entity | 1,000 | 27,734 |
| WebQSP | Entity/Number | 1,639 | 3,098 |
| GrailQA* | Entity/Number | 1,000 | 44,337 |
| Simple Quesiton* | Entity/Number | 1,000 | 14,894 |
| WebQuestions | Entity/Number | 2,032 | 3,778 |

**Table 8: Examples of faithful and interpretable reasoning of PoG.**

| Field | Content |
| --- | --- |
| Question | Lou Seal is the mascot for the team that last won the World Series when? |
| Answer | 2014 World Series |
| Topic Entity | **Lou Seal** |
| LLM indicator | "**Lou Seal**" - mascot for - team - last won - World Series - when |
| Split Question | split_question1: What team is "Lou Seal" the mascot for? |
| | split_question2: When did the last World Series win occur for the team with Lou Seal as the mascot? |
| Reasoning Path | {**Lou Seal**} → sports.mascot.team → {San Francisco Giants} → sports.sports_team.championships → {**2014 World Series**, 2012 World Series, 2010 World Series} |
| PoG answer | **answer**: {**2014 World Series**} |
| | **reason**: From the given path {Lou Seal} → sports.mascot.team → {San Francisco Giants}, {San Francisco Giants} is the team that "Lou Seal" serves as the mascot for. From {San Francisco Giants} → sports.sports_team.championships → {2014 World Series, 2012 World Series, 2010 World Series}, the last World Series won by the {San Francisco Giants} is in {2014}. Therefore, the provided knowledge graph path is sufficient to answer the overall question, and the answer is {2014 World Series}. |
| Question | What European Union country sharing borders with Germany contains the Lejre Municipality? |
| Answer | Denmark |
| Topic Entity | **{Germany, Lejre Municipality}** |
| LLM indicator | "**Lejre Municipality**" – contained by - European Union country (answer)- shares borders with- "**Germany**" |
| Split Question | split_question1: What European Union country shares borders with "Germany"? |
| | split_question2: What country contains the Lejre Municipality? |
| Reasoning Path | {**Lejre Municipality**} → location.administrative_division.country → {**Denmark**} → location.location.containedby → {Eurasia} → location.location.contains → {**Germany**}, {**Lejre Municipality**} → location.administrative_division.country → {**Denmark**} → base.locations.countries. continent → {Europe} → location.location.contains → {Western Europe} → location.location.contains → {**Germany**} |
| PoG answer | **answer**: {**Denmark**} |
| | **reason**: From {Denmark} → location.location.containedby → {Eurasia} →location.location.contains → {Germany}, Denmark is European Union country shares borders with "Germany". From {Lejre Municipality} → location.administrative_division.country → {Denmark}, Denmark is country contains the Lejre Municipality Both paths lead to Denmark, which is the European Union country sharing borders with Germany and containing the Lejre Municipality. Therefore, the provided knowledge graph path is sufficient to answer the main question, and the answer is {Denmark}. |
| Question | What Central Tiem Zone continent do the Falkland Islands belong to? |
| Answer | Americas |
| Topic Entity | **{Falkland Islands, Continent, Central Time Zone}** |
| LLM indicator | answer("**Central Time Zone**") – owns - answer("**Continent**")- located contain -"**Falkland Islands**" |
| Split Question | split_question1:What continent do the Falkland Islands belong to? |
| | split_question2: What Central Time Zone does the continent belong to? |
| Reasoning Path | {**Central Time Zone**} → time.time_zone.locations_in_this_time_zone → {**Americas**} → common.topic.not able_types → {**Continent**} → common.topic. notable_types → {Americas} → location.location.con tains →{ **Falkland Islands**} |
| PoG answer | **answer**: {**Americas**} |
| | **reason**: From {Continent} → common.topic.notable_types → {Americas} → location.location. contains →{ Falkland Islands}, Americas is the continent Falkland Islands belong to. From {Central Time Zone} → time.time_zone.locations_in_this_time_zone → {Americas} → common.topic. notable_types → {Continent}. Americas is the Central Time Zone. Therefore, the provided knowledge graph path is sufficient to answer the overall question, and the answer is {Americas}. |

## D   SPARQL

This section outlines the pre-defined SPARQL queries used for interacting with the knowledge graph and constructing graphs for our experiments.

### D.1   1-hop Entity and Relation Search

To facilitate the retrieval of 1-hop neighbors of entities within the Freebase Knowledge Graph, we have predefined a SPARQL query. This query is designed to be executed by simply substituting the appropriate ID for the query entity ID. It returns the connected entities' IDs and their associated relations' IDs, indicating whether the connected entity is at the tail or the head of the relation.

```
PREFIX ns: <http://rdf.freebase.com/ns/>
SELECT ?relation ?connectedEntity ?direction
WHERE {
    {
        ns:ID ?relation ?connectedEntity .
        BIND("tail" AS ?direction)
    }
    UNION
    {
        ?connectedEntity ?relation ns:ID .
        BIND("head" AS ?direction)
    }
}
```

### D.2   Short Textual Description

The following predefined function implements the retrieval of short textual descriptions, $\mathcal{D}(.)$, for converting the identifiers (IDs) of entities or relations into natural language descriptions.

```
PREFIX ns: <http://rdf.freebase.com/ns/>
SELECT DISTINCT ?tailEntity
WHERE {
    {
        ?entity ns:type.object.name ?tailEntity .
        FILTER(?entity = ns:ID)
    }
    UNION
    {
        ?entity <http://www.w3.org/2002/07/
            owlsameAs> ?tailEntity .
        FILTER(?entity = ns:ID)
    }
}
```

### D.3   1-hop Subgraph Search

To facilitate subgraph detection in Section 4.1, we implement the 1-hop subgraph detection feature by integrating SPARQL functions described in Appendix D.1 and D.2. The purpose of this function is to retrieve, in a single SPARQL query, the function returns the 1-hop neighbors of a given query with their IDs, natural language names, and connected relationships, specifying whether the connected entity is at the tail or the head of the relationship.

```
PREFIX ns: <http://rdf.freebase.com/ns/>
SELECT ?relation ?connectedEntity ?connectedEntityName ?
    direction
WHERE {
    {
        ns:ID ?relation ?connectedEntity .
        OPTIONAL {
            ?connectedEntity ns:type.object.name ?
                name .
            FILTER(lang(?name) = 'en')
        }
        BIND(COALESCE(?name, "Unnamed
            Entity") AS ?connectedEntityName)
        BIND("tail" AS ?direction)
    }
    UNION
    {
        ?connectedEntity ?relation ns:ID .
        OPTIONAL {
            ?connectedEntity ns:type.object.name ?
                name .
            FILTER(lang(?name) = 'en')
        }
        BIND(COALESCE(?name, "Unnamed
            Entity") AS ?connectedEntityName)
        BIND("head" AS ?direction)
    }
}
```

## E    PROMPTS

In this section, we detail the prompts required for our main experimental procedures.

---

**Question Analysis Prompt Template**

You will receive a multi-hop question, which is composed of several interconnected queries, along with a list of topic entities that serve as the main keywords for the question. Your task is to break the question into simpler parts, using each topic entity once and provide a Chain of Thought (CoT) that shows how the topic entities are related. Note: Each simpler question should explore how one topic entity connects to others or the answer. The goal is to systematically address each entity to derive the final answer.

In-Context Few-shot

Q: {Query}
Topic Entity: {Topic Entity}
A:

---

**LLM Supplement Prompt Template**

Using the main question, a possibly uncertain chain of thought generated by a language model, some related split questions, paths from the "Related_paths" section, and main topic entities: please first provide three predicted results, and second offer three possible chains of thought that could lead to these results, using the provided knowledge paths and your own knowledge. If any answers are unclear, suggest alternative answers to fill in the gaps in the chains of thought, following the same format as the provided examples.

In-Context Few-shot

Q: {Query}
Topic Entity: {Topic Entity}
Think Indicator:{Think Indicator}
Split Question:{Split Question}
A:

---

where {Think Indicator}, and {Split Question} are obtained in section 4.1. An indicator example is shown in Figure 2.

---

**Precise Path Select Prompt Template**

Given a main question, a LLM-generated thinking Cot that considers all the entities, a few split questions that you can use stepply and finally obtain the final answer, and the associated retrieved knowledge graph path, {set of entities (with id start with "m.")} -> {set of relationships} -> {set of entities(with id start with "m.")}, Please score and give me the top three lists from the candidate paths set can be highly to be the answer of the question.

In-Context Few-shot

Q: {Query}
Think Indicator:{Think Indicator}
Split Question:{Split Question}
Candidate Paths:{Candidate Paths}
A:

---

{Candidate Paths} denotes the retrieved reasoning paths $Final_paths$ to be selected in this request which are formatted as a series of structural sentences:

$$\{e_{0x}, ..., e_{0z}\} \rightarrow r_{1_i} \rightarrow \{e_{1x}, ..., e_{1z}\} \rightarrow ... \rightarrow r_{l_j} \rightarrow \{e_{lx}, ..., e_{lz}\}$$
$$...$$
$$\{e_{0x}, ..., e_{0z}\} \rightarrow r_{1_i} \rightarrow \{e_{1x}, ..., e_{1z}\} \rightarrow ... \rightarrow r_{l_j} \rightarrow \{e_{lx}, ..., e_{lz}\},$$

where $i$ and $j$ in $r_{1_i}, r_{1_i}$ represent the $i$-th, $j$-th relation from each relation edge in the clustered question subgraph. And $e$ is constructed by its ID and natural language name $\mathcal{D}(ID)$.

---

**Path Summarizing Prompt Template**

 Given a main question, an uncertain LLM-generated thinking Cot that consider all the entities, a few split questions that you can use stepply and finally obtain the final answer, the associated accuracy retrieved knowledge paths from the Related paths section, and main topic entities. Your task is to summarize the provided knowledge triple in Related paths section and generate a chain of thoughts by the knowledge triple related to the main topic entities of question, which will used for generating the answer for the main question and split question further. You have to make sure you summarize correctly by use the provided knowledge triple, you can only use the entity with id from the given path and you can not skip in steps.

In-Context Few-shot

Q: {Query}
Think Indicator:{Think Indicator}
Split Question:{Split Question}
Related Paths:{Related Paths}
A:

---

{Related_Paths} has the same format with the {Candidate_Paths} before.

---

**Question Answering Evaluation Prompt Template**

 Given a main question, an uncertain LLM-generated thinking Cot that considers all the entities, a few split questions that you can use and finally obtain the final answer, and the associated retrieved knowledge graph path, {set of entities (with id start with "m.")} -> {set of relationships} -> {set of entities(with id start with "m.")}. Your task is to determine if this knowledge graph path is sufficient to answer the given split question first then the main question. If it's sufficient, you need to respond {Yes} and provide the answer to the main question. If the answer is obtained from the given knowledge path, it should be the entity name from the path. Otherwise, you need to respose {No}, then explain the reason.

In-Context Few-shot

Q: {Query}
Think Indicator:{Think Indicator}
Split Question:{Split Question}
Related Paths:{Related Paths}
A:

---

**Question Answering Generation Prompt Template**

 Given a main question, an uncertain LLM-generated thinking Cot that consider all the entities, a few split questions that you can use stepply and finally obtain the final answer, and the associated retrieved knowledge graph path, {set of entities (with id start with "m.")} -> {set of relationships} -> {set of entities(with id start with "m.")}, Your task is to generated the answer based on the given knowledge graph path and your own knowledge.

In-Context Few-shot

Q: {Query}
Think Indicator:{Think Indicator}
Split Question:{Split Question}
Related Paths:{Related Paths}
A:

