# OpenReview forum: "Paths-over-Graph: Knowledge Graph Enpowered Large Language Model Reasoning"
_ACM.org/TheWebConf/2025/Conference — WWW 2025 Poster_

### Official Review · Reviewer_qenB · 2024-11-14

**Novelty:** 5
**Technical Quality:** 5

**Review:**

The paper proposes the Paths-over-Graph (PoG) method, which enhances the LLMs in complex, knowledge-intensive tasks through a three-stage dynamic multi-hop path exploration. The method demonstrates strong performance, particularly in multi-hop reasoning and multi-entity problems, while effectively leveraging graph structures.

### **Strengths:**
1. The experimental results are solid, providing strong support for the paper's claims.

2. The exploration of using knowledge graphs to enhance reasoning and reduce hallucinations is an important topic, and the paper makes meaningful contributions by addressing gaps in existing work.

### **Weaknesses:**
1. The paper analyzes the optimization of LLM token usage with PoG and compares the LLM calls counts of PoG and ToG in the appendix, demonstrating PoG's efficiency and cost-saving characteristics. However, it does not provide a direct comparison in terms of processing time—i.e., the average time required to process a given problem. This is relevant since many of PoG's modules seem to lack parallel processing capabilities, which could impact time efficiency.

**Questions:**

Could the authors provide a time-based comparison of PoG and other baselines, specifically the average time required to process a given problem?

**Reviewer Confidence:**

3: The reviewer is confident but not certain that the evaluation is correct

**Scope:**

4: The work is relevant to the Web and to the track, and is of broad interest to the community

---

### Official Review · Reviewer_LUS8 · 2024-11-30

**Novelty:** 4
**Technical Quality:** 4

**Review:**

This paper introduces Paths-over-Graph (PoG), a framework that combines the strengths of structured knowledge graphs (KGs) and the reasoning capabilities of LLMs to tackle QA tasks. Specifically, PoG exploits KG reasoning traces as multi-hop reasoning paths, effectively preserving the structural information of KG-retrieved knowledge. PoG integrates numerous techniques, including question decomposition, path pruning, and LLM summarization, to enhance QA performance. PoG achieves new SOTA results on multiple KGQA benchmarks.

Strengths:

- This paper introduces a novel KG-augmented LLM reasoning framework with a technically-sound approach. The authors present their motivation and methodology clearly throughout the paper.
- The authors conduct extensive experiments and analysis to demonstrate the effectiveness of their method.
- PoG achieves significant improvements over the previous SOTA method ToG.

Weeknesses:

- One of the core motivations claimed by the authors to integrate KGs into the LLM reasoning process is that updating LLM knowledge is resource-intensive, while updating KGs is easier. However, maintaining accurate up-to-date KGs can also be extremely costly, as it requires strong human supervision to ensure the structure and content of the updated knowledge are correct.
- While the proposed method PoG yields better performance than previous KG-based methods, the authors do not compare PoG with any SOTA text-based RAG methods to demonstrate the necessity of adapting KG knowledge for LLM QA.
- The paper shows limited relevance to the Web, with no explanation of how this work contributes to the Web community

**Questions:**

- For the “LLM supplement path exploration” step during the exploration phase, how is LLM hallucination prevented?
- Please double check your typos: title “Enpowered” should be “Empowered”, Figure 1(b) “Exlporated” should be “explored” or “exploited”, line 179 “planing” should be “planning” etc.

**Reviewer Confidence:**

3: The reviewer is confident but not certain that the evaluation is correct

**Scope:**

3: The work is somewhat relevant to the Web and to the track, and is of narrow interest to a sub-community

---

### Official Review · Reviewer_4RBc · 2024-12-01

**Novelty:** 5
**Technical Quality:** 4

**Review:**

Quality:
The paper presents a framework for integrating Knowledge Graphs with LLMs for better reasoning, with a focus on multi-hop and multi-entity reasoning. The study is thorough and the framework is evaluated across several benchmarks and SOTA models. The results show a clear performance increase compared to existing models.

Clarity:
The paper addresses current challenges systematically. There are some figures (Figures 1 and 2) that, while complex, bring a lot of clarity to the content of the paper. Figure 3 might bring too many details for the main body of the paper. I suggest moving that to the Appendix.
The in-line formalization of the knowledge graph G(E,R,T) is at places difficult to read, particularly when it follows the keyword KG, which happens a few times throughout the paper. It might be helpful to consider changing the style of your in-line math for better readability.

Originality:
The paper introduces novelty through its intricate pipeline. While previous work on reasoning over Knowledge Graphs with LLMs exists, I believe this study tries to leverage as much information out of the structured data as possible. Additionally, the proposed method aims to improve the faithfulness of reasoning by enhancing the interpretability and alignment of LLM outputs with the underlying knowledge graph structure.

Significance of work:
I believe this work is significant for the KG community

Strengths:
The submission has a lot of self-explanatory figures that help with understanding the content of the paper
The methodology is formalized properly. PoG handles complex multi-hop and multi-entity queries through a structured approach.
The examples of the reasoning and entity paths in Section 3 are nice for getting a better understanding of the methodology.
The method enhanced the interpretability and faithfulness of LLM reasoning over KGst
Interesting multi-hop and multi-entity schemes for dynamic path exploration

Weaknesses:
LLM impact: The pipeline still relies too much on LLMs, including question analysis, path exploration, path pruning, and final QA. The authors do acknowledge the issue of LLM hallucinations, however, the method still heavily relies on these models. This heavily impacts the robustness of the methodology.
LLM impact #2: In the Path Pruning stage, which relies on LLMs, the model will likely choose the LLM-generated path especially because it has been self-generated (LLMs perform better when they are familiar with the input, which happens when it is self-generated). Therefore, the bias is even higher. This heavily impacts the robustness of the methodology.
Section 4 Initialization: When I first read this section, because you mentioned “from each topic entity to construct the evidence sub-graph” I thought there was one entity that was more “central” than the others. But later it became clearer that you retrieve a subgraph for each entity recognized in the question/query. I would make this more explicit in that section, as it introduces this stage of your methodology.
More experiments: it would be interesting to see how your method performs using the LLMs of the other SOTA approaches that use both LLMs and external knowledge. For example, I see TOG also uses Llama2-70B
The pipeline is complicated, including many substages in all of the main stages of the method.
Small detail: there might be a typo in your title, should it be Empowered?
Not a real weakness, but to note for the next iteration: the name of the conference should be changed in the template

**Questions:**

1. Related to Weakness 1: If I understand correctly, there is a part of the methodology that does 1) Initialization which contains Question Analysis using LLMs, 2) Exploration in 3 stages, with one being LLM supplement path exploration, 3) Path Pruning with 3 stage beam search, where one stage is LLM prompting, and 4) The QA is done also with LLM prompting. How can you ensure, given there are 3-4 subsequent stages using LLMs, that the reasoning is not heavily biased and flawed? You make a point that LLMs hallucinate, and while you only rely on LLMs as part of your pipeline, they can still heavily affect you through propagated hallucinations.
2. Related to Weakness 2: Do you have a way to mitigate this? Are you using the same LLM/different ones? You address the hallucination problem with Path Summarizing, however, can you ensure that is enough?
3. In Section 3, it is mentioned that given an entity set E_S, the induced subgraph is denoted as S = (E_S, R_S, T_S) where both the head and tail entities have to be in E_S. Is this something you wanted/desirable behaviour? My intuition is that maybe just one of them should be in E_S. Can you give any clarity on that?
4. You mention in the Initialization phase that you first retrieve a subgraph of depth D_max, and then you use the LLM to analyze the question and generate a depth indicator D_predict. You use these in the Exploration phase, with depth D in range [D_predict, D_max]; How do you ensure D_predict is smaller? Given that you don’t prompt the LLM with the subgraph.

**Reviewer Confidence:**

3: The reviewer is confident but not certain that the evaluation is correct

**Scope:**

3: The work is somewhat relevant to the Web and to the track, and is of narrow interest to a sub-community

---

### Official Review · Reviewer_Hzo5 · 2024-12-02

**Novelty:** 5
**Technical Quality:** 6

**Review:**

The paper proposes a KGQA framework and achieves SOTA across multiple datasets. This provides valuable insights and reference for future research in KGQA. It is well-written, making it easy to read, understand, and follow. It conducts experiments on multiple datasets, the method demonstrates outstanding performance, validating its effectiveness. However, the evaluation metric only utilizes Hits@1, lacking the use of more comprehensive metrics such as F1. Moreover, the datasets are not fully utilized, with only 1,000 samples being selected.

**Questions:**

1.During the question decomposition process, how are multi-hop questions decomposed into sub-questions? For instance, in the case of multi-hop questions, the generation of subsequent questions depends on the answers to preceding ones, making it impossible to generate all sub-questions at once.
2.The experiments exclusively utilize closed-source LLMs. Given the limitations of closed-source LLMs, such as privacy concerns and high costs, can this method be effectively applied to open-source LLMs?
3.Is the process of sampling reasoning paths from a knowledge graph overly complex? This includes steps like question subgraph detection, exploration, and path pruning.

**Reviewer Confidence:**

3: The reviewer is confident but not certain that the evaluation is correct

**Scope:**

4: The work is relevant to the Web and to the track, and is of broad interest to the community

---

### Official Review · Reviewer_jL6Y · 2024-12-03

**Novelty:** 3
**Technical Quality:** 4

**Review:**

This paper introduces “Paths-over-Graph” (PoG), a method that integrates Knowledge Graphs (KGs) and Large Language Models (LLMs) to enhance reasoning capabilities for multi-hop and multi-entity tasks. The approach combines structured knowledge from KGs with LLMs’ inherent abilities through a three-phase exploration and pruning strategy, aiming to improve interpretability, faithfulness, and efficiency in reasoning.

Strengths:
1. The paper identifies critical challenges in KG-augmented LLM reasoning, such as multi-hop reasoning and efficient graph pruning.
2. PoG introduces a three-phase exploration process that integrates LLM prompting with graph-structured pruning for efficient knowledge selection.
3. PoG achieves performance improvements over baseline methods across multiple datasets, particularly in multi-hop and multi-entity tasks.

Weaknesses
1. The core methodology of PoG lacks significant innovation compared to existing methods like ToG and ToG 2.0. The paper does not sufficiently clarify how PoG differs fundamentally or offers a breakthrough relative to these established approaches.
2. Figures 2 and 3 are densely packed with information but have minimal accompanying explanations in the captions, making them difficult for readers to comprehend.
3. While the paper discusses the pruning techniques and LLM token savings, it does not provide a comprehensive runtime or computational efficiency analysis of the method relative to other approaches.

While the paper addresses an important problem and shows promising results, the lack of clarity on its novelty, overly complex visuals, and insufficient analysis of efficiency hinder its overall contribution. Thus, I don't think the paper is ready for publication yet, and I encourage the authors to address these concerns in a revised submission.

**Questions:**

1. Can you clarify the computational efficiency of PoG compared to baseline methods other than #LLM calls (e.g., runtime, memory usage)? Are there practical trade-offs in implementing the proposed approach?

**Reviewer Confidence:**

3: The reviewer is confident but not certain that the evaluation is correct

**Scope:**

3: The work is somewhat relevant to the Web and to the track, and is of narrow interest to a sub-community